# Biglycan Alleviates Age-Related Muscle Atrophy and Hepatocellular Senescence

**DOI:** 10.3390/ijms26178286

**Published:** 2025-08-26

**Authors:** Da Som Lee, Joo Hyun Lim, Yoo Jeong Lee

**Affiliations:** Division of Endocrine and Kidney Disease Research, Department of Chronic Disease Convergence Research, National Institute of Health, Cheongju 28159, Republic of Korea; dasom12@korea.kr (D.S.L.); mikorio@korea.kr (J.H.L.)

**Keywords:** biglycan, BGN, myokine, muscle atrophy, older adults, senescence, exercise

## Abstract

Myokines are secreted by muscle and play crucial roles in muscle repair and regeneration and also impact diverse physiological effects through crosstalk with other metabolic organs. However, aging is associated with a progressive decline in muscle mass, which in turn leads to reduced myokine secretion. This decline may contribute to the development of sarcopenia, leading to an increased risk of metabolic disorders such as type 2 diabetes. Accordingly, interest in identifying novel myokines and elucidating their biological functions is increasing. In this study, we explored the function of biglycan (BGN), a novel myokine, in aging-related metabolic tissues. BGN levels decreased in the muscle tissue and plasma of older adults and aged mice, whereas exercise intervention restored BGN expression in aged mice. BGN counteracted the expression of atrophy-related genes involved in muscle degradation and mitigated muscle mass loss by regulating AKT/mTOR signaling pathway. Notably, BGN decreased the expression of the senescence marker p21 and senescence-associated secretory phenotype (SASP)-related genes in hepatocytes. Additionally, BGN attenuated senescence-induced lipid accumulation and ROS generation. Our results suggest that BGN has beneficial effects against muscle atrophy and hepatocellular senescence, indicating its potential as a protective factor for age-related diseases.

## 1. Introduction

Changes in the mass and functional capacity of skeletal muscle are closely associated with aging progression [1]. Recent evidence has indicated that skeletal muscles, as secretory organs, generate and release myokines that influence systemic energy metabolism and glucose homeostasis through autocrine, paracrine, and endocrine pathways in response to physical exercise. Therefore, age-related declines in physical activity may lead to alterations in both the secretion and responsiveness of myokines, ultimately increasing susceptibility to sarcopenia and metabolic disorders associated with aging. Myokines serve as key mediators for facilitating communication between skeletal muscle and other metabolic tissues, including liver tissue, suggesting that the beneficial effects of exercise on various organs are attributed to the potential protective role of myokines in metabolic disease and aging. However, most secreted myokines are still not sufficiently characterized.

The skeletal muscle extracellular matrix (ECM) plays crucial roles in muscle development, maintenance, repair, and atrophy. Culture on ECM extracted from the large thigh muscles of rats promoted myogenic cell proliferation and differentiation ex vivo, suggesting extensive interactions between the ECM and cells [2]. Biglycan (BGN), a member of the small leucine-rich proteoglycan (SLRP) family, is primarily responsible for organizing the ECM and has been shown to be involved in skeletal muscle development and regeneration after muscle damage is induced [3]. BGN functions as a regulator of the expression and sarcolemmal localization of dystrobrevin, syntrophin, and nNOS; therefore, BGN-null mice exhibit a dystrophic phenotype and increased muscle fiber degeneration [4]. Moreover, the knockdown or absence of *BGN* impairs healing and alters mechanical functions in aged tendons, indicating that BGN plays a critical role in maintaining tissue homeostasis in aged tendons [5,6]. However, these studies focused primarily on the intrinsic ECM mechanical properties of BGN.

Recent studies have provided mechanical insights into the role of BGN as a potential myokine with regulatory functions beyond its structural role [7,8]. Hartwig et al. identified 305 proteins classified as potential myokines from the secretome of primary human skeletal muscle cells (hSkMCs), among which BGN was included as a myokine with a signal peptide secreted by hSkMCs [9]. Additionally, BGN serves as a ligand for key signaling pathways, including the insulin-like growth factor I receptor (IGF-IR) pathway [10]. BGN facilitates the nuclear translocation of IGF-IR by directly binding to it, thereby sustaining IGF-IR activation and influencing target gene regulation. This process is linked to increased cellular proliferation and resistance to stress conditions in certain cell types. Emerging evidence suggests that BGN functions as a signaling molecule involved in metabolic regulation. It has been demonstrated that the administration of BGN reduces body weight and improves glucose uptake in the skeletal muscles of high-fat diet (HFD)-induced obese mice [11].

In this study, we identified BGN as a novel myokine and examined the impact of BGN on age-associated disease. Although BGN has been implicated in the regulation of metabolic mechanisms, its precise functions during aging remain unexplored. Our findings demonstrate that BGN derived from skeletal muscle may act as a positive effector of aging-related muscle atrophy. We further showed that BGN serves important roles as a mediator of crosstalk between skeletal muscle and the liver by alleviating senescence-associated hepatic steatosis.

## 2. Results

### 2.1. BGN Expression Progressively Declines with Age

To determine the age-related changes in *BGN* levels, we reanalyzed transcriptome data obtained from the Gene Expression Omnibus (GEO) database [12]. Analysis of the GSE175495 dataset revealed that *BGN* expression in skeletal muscle was significantly lower in older human subjects than in young subjects and that the expression of forkhead box O3 (*Foxo3*), which plays a pivotal role in muscle atrophy, was increased in older subjects (Figure 1A). In line with these results, the BGN level in human plasma gradually decreased with age, according to proteomic data reported by Lehallier et al. (Appendix A) [13]. In an aging mouse model, which was used in our previous study [14], the mRNA and protein expression of BGN was lower in the gastrocnemius (GA) muscle of aged mice than in that of young mice but was effectively rescued by exercise training (Figure 1B,C). We subsequently generated an accelerated aging mouse model using doxorubicin (Dox) (Figure 1D). The expression of senescence markers (*p16* and *p21*), SASP markers (*IL6*, *Ccl2*, and *Tgfβ1*), and protein degradation-related genes (*Foxo3*, *4EBP1*, and *Atrogin*) increased in the GA muscle of Dox-treated mice (Figure 1E), indicating the induction of senescence in muscle. Consistent with these findings, we observed a significant reduction in BGN levels in both the plasma and GA muscles of Dox-treated mice compared with control mice (Figure 1F,G). Collectively, these findings indicate that the expression of *BGN*, which serves as a myokine, is downregulated during aging but upregulated in response to exercise.

### 2.2. BGN Is a Secreted Factor Associated with Muscle Function

We subsequently assessed the expression and secretion of BGN during myogenesis, which is crucial for muscle formation, regeneration, and growth. During the differentiation of C2C12 myoblasts into myotubes, the expression of myogenin, myosin heavy chain (MyHC), and desmin, which are markers of myogenesis, increased upon the induction of differentiation. Similarly, the abundance of BGN increased on day 2 and gradually increased over 6 days during the process of myogenic differentiation (Figure 2A,B). Moreover, the secretion of BGN was greater in fully differentiated myotubes than in myoblasts (Figure 2C). To provide additional evidence of exercise-induced myokines, we analyzed BGN levels in myotubes following exercise-mimic stimulation. Fully differentiated myotubes were treated with forskolin (Fsk), which promotes cAMP pathway activation via the activation of adenylate cyclase activity. As anticipated, the mRNA expression of *BGN* and its level of secretion into the culture medium markedly increased after Fsk treatment (Figure 2D,E). These results suggest that BGN is released from mature myotubes and that BGN secretion increases in response to exercise stimulation.

### 2.3. BGN Has a Protective Effect Against Muscle Atrophy

Muscle atrophy is defined as a decline in muscle mass and function and is closely correlated with aging; thus, we next examined the effect of BGN on muscle atrophy. As observed in aged mice, dexamethasone (Dex), a glucocorticoid known to cause muscle atrophy by triggering the catabolism of muscle proteins, induced the expression of atrophy-related genes such as *Atrogin* and *MuRF1*, whereas the expression of *BGN* was significantly reduced in Dex-induced C2C12 myotube atrophy (Figure 3A). Importantly, biglycan treatment restored the thickness and length of myotubes impaired by Dex, as shown by MyHC staining (Figure 3B,C). Accordingly, BGN diminished the Dex-induced expression of muscle protein degradation-related genes, including *Atrogin*, myostatin (*MSTN*), 15-hydroxyprostaglandin dehydrogenase (*Hpgd/15-PGDH*), Krüppel-like factor 15 (*Klf15*), and *Foxo3*, in C2C12 myotubes (Figure 3D,E). These data indicate that BGN effectively alleviates Dex-induced muscle atrophy.

### 2.4. BGN Promotes the AKT/mTOR Pathway to Increase Protein Synthesis

We then investigated the signaling mechanism through which BGN modulates muscle atrophy. Given that aging-related muscle atrophy results from increased muscle protein breakdown and decreased synthesis of new proteins [1], we hypothesized that BGN promotes protein synthesis. Since the AKT/mTOR pathway plays a key role in promoting protein synthesis in skeletal muscle, we assessed the activation of AKT/mTOR signaling. Western blot data revealed that the phosphorylation levels of serine-threonine kinase (AKT), mammalian target of rapamycin (mTOR), 70-kDa ribosomal protein S6 kinase (p70S6K), and eukaryotic translation initiation factor 4E-binding protein 1 (4EBP1), critical regulators of the translational machinery involved in protein synthesis, significantly increased in C2C12 myotubes treated with BGN (Figure 4A,B). Following these findings, we further examined the effect of BGN on protein synthesis rates and revealed increased puromycin incorporation into newly generated proteins under BGN-treated conditions (Figure 4C). Consistent with these findings, we observed a significant reduction in the phosphorylation levels of p70S6K and 4EBP1 after Dex treatment, whereas these adverse effects on muscle protein synthesis were abolished by treatment with BGN (Figure 4D). Collectively, these results demonstrate that BGN ameliorates muscle atrophy by promoting protein synthesis via activation of the AKT/mTOR axis.

### 2.5. Dox-Induced Hepatocellular Senescence Is Restored by BGN

During human aging, the liver undergoes various physiological and cellular changes that can increase susceptibility to liver damage and disease. Specifically, aging is associated with increased risk of liver diseases, including nonalcoholic fatty liver disease (NAFLD) and fibrosis [15]. Considering that BGN secretion decreases in older adults but increases through exercise, we next sought to determine whether BGN, as a myokine, plays a functional role in regulating the senescence of other metabolic tissues, especially the liver, beyond skeletal muscle. To this end, AML-12 cells, a normal mouse hepatocyte line, were treated with various doses of Dox, and 0.2 µM Dox was confirmed as an appropriate dose on the basis of cell viability (Figure 5A) as well as significant senescence-related changes in the expression of p21 and phosphorylated histone H2AX (γH2AX) (Figure 5B,C). Similarly, BGN treatment markedly decreased the number of SA-β-galactosidase-positive cells (Figure 5D,E) and suppressed the mRNA levels of *p21*, *IL6*, and *TNFα*, which are senescence-associated secretory phenotype (SASP)-related genes (Figure 5F), in Dox-treated hepatocytes. In addition, the protein level of p21 decreased, whereas that of the proliferation marker Ki67 was restored after treatment with BGN (Figure 5G). These findings indicate that BGN attenuates cellular senescence in hepatocytes.

### 2.6. BGN Inhibits Lipid Accumulation During Hepatic Senescence

Given that hepatocyte senescence is closely linked to intracellular lipid accumulation, we next evaluated the effect of BGN on aging-associated lipid accumulation in hepatocytes. BGN ameliorated hepatic triglyceride (TG) accumulation induced by palmitic acid (PA) and Dox in cells (Figure 6A,B). Consistent with these observations, the expression of lipogenic genes, including fatty acid translocase (*FAT/CD36*), glycerol-3-phosphate acyl-transferase (*GPAT*), and *lipin1*, was reduced in hepatocytes treated with BGN (Figure 6C). Oxidative stress is among the major mechanisms that lead to hepatic steatosis during aging [16]. In senescent cells, mitochondrial dysfunction continuously generates reactive oxygen species (ROS), which contribute to the impairment of metabolic function [17]. As expected, we observed elevated ROS levels in Dox-treated hepatocytes, but these changes were markedly reversed by BGN (Figure 6D,E). In accordance with the increase in oxidative stress, the expression of the antioxidant genes NAD(P)H quinone dehydrogenase 1 (*Nqo1*), heme oxygenase 1 (*Hmox1*), and catalase (*CAT*) decreased in Dox-treated cells (Figure 6F). These data suggest that the antisenescent effect of BGN on hepatocytes consequently attenuates aging-induced hepatic TG accumulation and oxidative stress.

Collectively, our results indicate that BGN expression is upregulated by exercise and acts in an autocrine manner to preserve muscle mass by enhancing protein synthesis via the AKT/mTOR pathway, thereby alleviating skeletal muscle atrophy. In the liver, BGN potentially plays a role in regulating cellular senescence, age-related lipid accumulation, and the SASP (Figure 7).

## 3. Discussion

Aging impairs the physical and metabolic functions of skeletal muscles, leading to diminished physical activity in daily life and increased risks of age-related disease. Among the age-related physiological changes, sarcopenia, characterized by the loss of muscle mass, is a major pathological feature of aging. Accumulating evidence suggests that muscle aging affects not only structural integrity but also various physiological functions, including impaired metabolic homeostasis, mitochondrial dysfunction, and global alterations in transcriptomic and proteomic profiles [18,19,20].

Age-related alterations in muscle tissue are closely associated with changes in molecular signaling factors such as growth factors and hormones, especially the altered expression of myokines secreted from skeletal muscle. For this reason, we aimed to identify a myokine for which expression decreases with increasing muscle age but increases with exercise, thereby contributing to the restoration of muscle function. By performing a comparative analysis of muscle-derived secretory proteins between young and older subjects, we identified BGN as a myokine with markedly reduced expression in older individuals. To date, BGN has been studied primarily as a central structural component of the ECM, contributing to the reinforcement of the musculoskeletal system, and its deficiency has been associated with musculoskeletal abnormalities [21]. In this study, we demonstrate a previously unrecognized role of BGN as a myokine derived from skeletal muscle that ameliorates age-related changes in muscle. Importantly, BGN expression was significantly decreased in the muscle tissue and plasma of aged mice; however, exercise intervention restored its expression. We also found that exercise-mimicking stimulation in vitro enhanced the release of BGN into the culture medium. In agreement with our study, a transcriptomic analysis of human skeletal muscle revealed an increase in BGN following exercise, and circulating levels of BGN were also increased in exercised rodents and humans [11,22,23]. These results suggest that BGN may serve as an exerkine, mediating the effect of exercise.

During aging, skeletal muscle displays decreased protein synthesis and upregulated atrophy-related gene expression, reflecting a shift toward catabolic metabolism [24]. The suppression of the AKT/mTOR signaling pathway, a major anabolic pathway that controls muscle mass, is considered a critical contributor to the progression of sarcopenia [25,26]. Here, we observed that BGN suppressed the Dex-induced expression of proteolysis-related genes and promoted protein biosynthesis by activating the AKT/mTOR signaling pathway, consequently mitigating atrophy in myotubes. Therefore, our data suggest that BGN may protect muscle fibers under stress conditions by modulating anabolic and catabolic signals. Notably, exercise is known to trigger the AKT/mTOR pathway to induce muscle hypertrophy [27]. Our results indicate that BGN, as an exerkine, may partially mediate the benefit of exercise with respect to muscle adaptation.

Myokines act via autocrine and paracrine mechanisms to regulate energy homeostasis, regeneration, and inflammation in muscles. Recent studies have highlighted the endocrine role of myokines, in addition to their mechanical functions; the endocrine aspects of myokines, in turn, influence systemic metabolic functions in other organs. In relation to metabolic crosstalk between skeletal muscle and bone, cardiotrophin-like cytokine factor 1 (CLCF1), which is derived from muscle in response to exercise, has been reported to alleviate age-related functional decline in both muscle and bone [28]. Some myokines, including irisin and meteorin-like (Metrnl), play roles in regulating lipid metabolism and inducing the browning of white adipose tissue involved in mechanisms associated with obesity and metabolic syndrome, such as the beneficial effect of physical exercise [29]. Additionally, thrombospondin-4 (Thbs4), a novel myokine produced in skeletal muscle during hypertrophy and aerobic exercise, reprograms the differentiation of the stromal vascular fraction to generate thermogenic adipocytes in subcutaneous adipose tissue [30].

The prevalence of NAFLD and NASH tends to increase with advancing age. With age, the ability of the liver to oppose stress decreases. Specifically, aging increases susceptibility to liver injury because of elevated oxidative stress and inflammatory responses, accelerated cellular senescence, and impaired mitochondrial function. However, exercise and dietary interventions in elderly individuals have been reported to reduce hepatic lipid accumulation and improve insulin resistance, likely because of the effects of exercise-induced myokines on the aging liver [31]. In line with previous findings, our conclusions revealed that BGN treatment downregulated the expression of the senescence marker p21 and decreased the number of senescent hepatocytes; additionally, it mitigated lipid accumulation and ROS generation induced by Dox and PA, thereby enhancing the resistance of the cells to stress, indicating that BGN serves as an antisenescent factor. These results support the hypothesis that exercise-induced BGN counteracts oxidative stress and fat accumulation-induced hepatocyte senescence via myokine-mediated skeletal muscle-liver crosstalk.

In summary, our study identified a critical role for BGN in regulating aging-associated metabolic pathways and functional decline in muscle cells and hepatocytes. Despite these novel findings, a limitation of this study is the lack of in vivo validation. Further in vivo studies should be performed using muscle-specific BGN deletion or administration models to elucidate the physiological functions of BGN during aging. In addition, large-scale analyses of human clinical cohorts are needed to determine the associations between circulating BGN levels and aging-related disorders. The results of this study indicate that BGN has the potential to serve as a new protective factor against age-associated diseases and metabolic stress.

## 4. Materials and Methods

### 4.1. Animal Experiments

All mouse experiments and procedures were approved and conducted in accordance with the guidelines of the Institutional Animal Care and Use Committee of Korea National Institute of Health (permit numbers: KCDC-032-20-2A and KDCA-IACUC-24-030). The experimental procedures used for the aging mouse model have been thoroughly described previously [14]. Briefly, aged (18-month-old) C57BL/6 male mice (n = 10–12 per group) were randomly divided into 2 groups: a control group and an exercise group. The exercise intervention was performed 3 days a week for 4 months using a treadmill (Panlab, Cornellà de Llobregat, Spain). During the first week, the mice were allowed to adapt to the treadmill. After adaptation training, the mice were allowed to run at 5 m/min for 5 min and then at 13 m/min for 15 min/day 3 days a week for 4 months. For Dox-induced aging mouse experiments [32], a schematic diagram of the mouse model of Dox-induced aging is shown in Figure 1D. Eight-week-old male C57BL/6 mice were purchased from Kone Biotech (Seoul, Republic of Korea) and acclimatized for one week prior to the experiments. Dox was injected intraperitoneally at a dosage of 5 mg/kg every 3 days for 2 weeks, and PBS was injected as a negative control (n = 5 per group). After 2 weeks, the mice were euthanized, and the harvested tissues were frozen immediately. All the mice were maintained on a 12 h light/dark cycle under temperature- and humidity-controlled conditions and were allowed to acclimate for one week with free access to food and water.

### 4.2. Cell Culture and Reagents

C2C12 myoblasts (ATCC, Manassas, VA, USA) were maintained in Dulbecco’s modified Eagle’s medium (DMEM; Gibco, Waltham, MA, USA) supplemented with 10% fetal bovine serum (FBS; Gibco, Waltham, MA, USA) and 1% penicillin/streptomycin (Gibco, Waltham, MA, USA). To induce myogenic differentiation, the growth medium was changed to DMEM supplemented with 2% horse serum (Gibco, Waltham, MA, USA) when the cells reached 80–90% confluence. AML12 mouse normal hepatocytes (ATCC, Manassas, VA, USA) were cultured in DMEM/F12 (Gibco, Waltham, MA, USA) supplemented with 10% FBS, 1% penicillin/streptomycin, 5 μg/mL insulin, 5 μg/mL transferrin, 5 ng/mL selenium, and 40 ng/mL dexamethasone. Doxorubicin (Dox, D1515), dexamethasone (Dex, D4902), and palmitic acid (PA, P5585) were purchased from Sigma-Aldrich (Saint Louis, MO, USA). BGN (8128-CM-050) was purchased from R&D Systems (Minneapolis, MN, USA).

### 4.3. Quantitative PCR

Total RNA from cells or mouse skeletal muscle tissue was isolated using an RNeasy Mini Kit (QIAGEN, Germantown, MD, USA). Complementary DNA (cDNA) was transcribed from 2 µg of total RNA using SuperScript III Reverse Transcriptase (Invitrogen; Thermo Fisher Scientific, Waltham, MA, USA). Quantitative PCR (qPCR) was performed using SYBR Green Master Mix (Thermo Fisher Scientific, Waltham, MA, USA) and a QuantStudio 6 Flex system (Thermo Fisher Scientific, Waltham, MA, USA). All the data were normalized to the expression of *GAPDH*, *Hprt*, or *L32*. The primer sequences are listed in Appendix A.

### 4.4. Immunoblotting

Cells and muscle tissue were homogenized in RIPA buffer supplemented with protease inhibitors (Sigma-Aldrich, St. Louis, MO, USA). Total protein was quantified using a BCA protein assay (Pierce; Thermo Fisher Scientific, Waltham, MA, USA), separated by electrophoresis on SDS–polyacrylamide gels, and transferred to polyvinylidene difluoride (PVDF) membranes. The membranes were subsequently blocked with 5% skim milk and probed with primary antibodies. Antibodies against BGN and γH2AX were obtained from Abcam (Cambridge, UK). Antibodies against AKT, phospho-AKT, mTOR, phospho-mTOR, S6 kinase (S6K), phospho-S6K, 4EBP1, phospho-4EBP1, S6, phospho-S6, Foxo3, p21, Ki67, and β-actin were obtained from Cell Signaling Technology (Danvers, MA, USA). Antibodies against MyoD, MyHC, Atrogin, and MuRF were purchased from Santa Cruz (Santa Cruz, CA, USA).

### 4.5. Measurement of BGN

To assess BGN secretion from myoblasts and myotubes, the culture medium was changed with fresh serum-free medium. After a 1-h incubation, the medium was harvested and centrifuged at 1300 rpm for 5 min at 4 °C, and then the supernatant was collected. BGN levels were subsequently determined using a Mouse Biglycan ELISA Kit (Abcam, Cambridge, UK, ab275906) in accordance with the manufacturer’s instructions.

### 4.6. Immunofluorescence Staining

Differentiated myotubes were fixed in 4% paraformaldehyde for 10 min and then were washed three times with PBS. After permeabilization with PBS containing 0.2% Triton for 5 min, the myotubes were blocked with 2% BSA for 1 h at 4 °C. The myotubes were subsequently incubated with an anti-Myosin 4 monoclonal antibody conjugated with Alexa Fluor 488 (1:200; Invitrogen, Waltham, MA, USA) in 0.2% BSA overnight at 4 °C. The next day, the myotubes were washed with PBS three times and incubated with DAPI. After five washes in PBS, images were acquired using a confocal microscope (Olympus, FV3000, Tokyo, Japan), and the diameter and length of the myotubes were quantified using ImageJ software, version 1.54p (NIH, Bethesda, MD, USA).

### 4.7. SUnSET Assay

To evaluate protein synthesis in C2C12 myotubes, on day 5 of differentiation, the myotubes were treated with 1 µM puromycin for 1 h. Following treatment, the myotubes were harvested and subjected to Western blotting to detect puromycin-incorporated peptides.

### 4.8. β-Galactosidase Staining

AML12 cells were seeded at a confluence of 60% and treated with Dox with or without 1 μg/mL BGN pretreatment for 24 h at the concentrations indicated in the figure legend, and PBS was used as a negative control. After treatment, the cells were rinsed with PBS and stained to identify senescent cells using a commercial kit (Cell Signaling Technology, Danvers, MA, USA) in accordance with the manufacturer’s protocol. Following staining, the cells were examined using a light microscope, and the β-galactosidase-positive areas were quantified with ImageJ software (NIH, Bethesda, MD, USA).

### 4.9. ROS Assay

AML12 cells were seeded in 6-well plates and treated with 0.2 µM Dox and 200 µM PA for 24 h with or without 1 μg/mL BGN pretreatment. A 2% BSA solution was used as a negative control. After treatment, the cells were washed twice with PBS and stained with 10 µM DCFDA (Abcam, Cambridge, UK, ab113851) for 45 min at 37 °C. Following staining, the intracellular ROS levels were observed using a fluorescence microscope (Olympus, Tokyo, Japan).

### 4.10. Statistical Analysis

All the data are presented as the means ± standard deviations (SD). Statistical analyses were performed using GraphPad Prism version 10 software (GraphPad, La Jolla, CA, USA). Comparisons between two groups were performed using Student’s *t* test or the nonparametric Mann–Whitney U test. For multiple-group comparisons, we determined whether the data from each group followed a normal distribution using the Shapiro–Wilk test and the Kolmogorov–Smirnov test. In the case of normality, we performed a parametric one-way ANOVA with a Tukey post hoc test for multiple comparisons to assess the significance among pairs of conditions; in the case of nonnormality, we performed a Kruskal–Wallis test and a nonparametric post hoc test. For representative data, at least two independent experiments were performed to ensure reproducibility. Values of *p* < 0.05 were considered to indicate statistical significance.

## Figures and Tables

**Figure 1 ijms-26-08286-f001:**
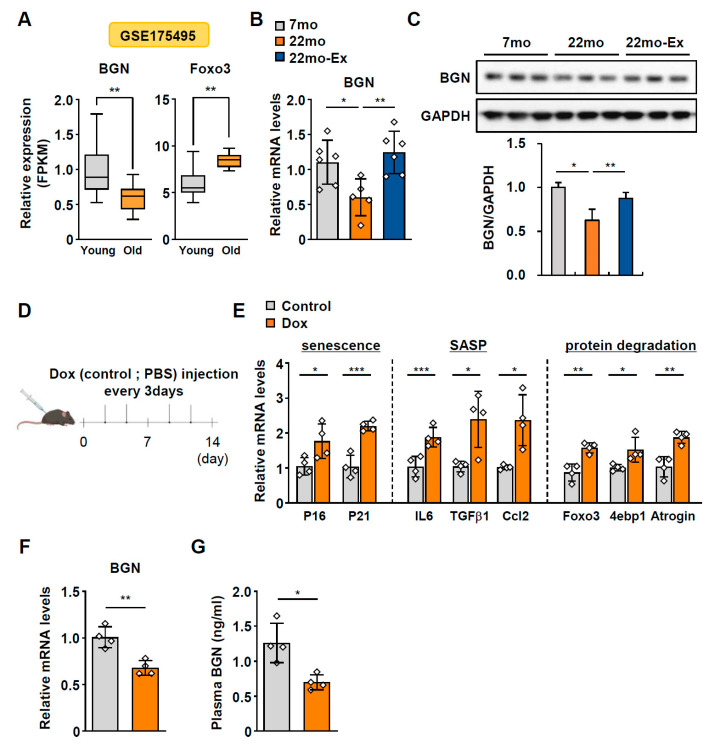
BGN decreases with age in the skeletal muscle of humans and mice. (**A**) Expression levels of biglycan (*BGN*) and *Foxo3* in the quadriceps muscle of young (n = 10) and older (n = 10) human subjects from the GSE175495 dataset. (**B**,**C**) Changes in the (**B**) mRNA (normalized to *Hprt*) and (**C**) protein levels of BGN in the gastrocnemius (GA) muscles of young (7-month-old) and older (22-month-old) mice fed a normal chow diet (ND) or ND plus exercise (exe) (n = 6–7 per group). (**D**) A schematic diagram (created using BioRender.com) showing the Dox-induced aging mouse model. (**E**) Relative mRNA levels (normalized to *GAPDH*) of senescence (*p16* and *p21*), atrophy (*Foxo3*, *4ebp1*, and *Atrogin*), and SASP (*IL6*, *Tgfβ1*, and *Ccl2*)-related genes in the GA muscle of control (PBS) and Dox-injected mice (n = 4 per group). (**F**) mRNA (normalized to *GAPDH*) and (**G**) plasma levels of BGN in control (n = 4) and Dox-treated mice (n = 4). The data are presented as the mean  ±  SD; * *p* < 0.05, ** *p* < 0.01, and *** *p* < 0.001.

**Figure 2 ijms-26-08286-f002:**
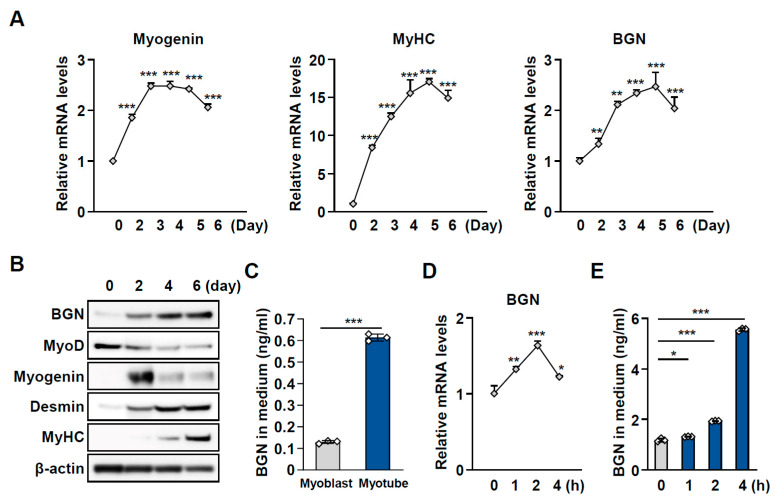
BGN expression is upregulated, and BGN is secreted throughout differentiation. (**A**) mRNA expression (normalized to *GAPDH*) and (**B**) immunoblotting of BGN and myogenic markers in C2C12 cells during myogenic differentiation and the corresponding quantification. (**C**) BGN concentration in the culture medium of C2C12 myoblasts and myotubes. (**D**) BGN mRNA expression and (**E**) BGN accumulation in the culture medium of C2C12 myotubes after treatment with 2 µM forskolin for the indicated times. The data are presented as the mean  ±  SD; * *p* < 0.05, ** *p* < 0.01, and *** *p* < 0.001.

**Figure 3 ijms-26-08286-f003:**
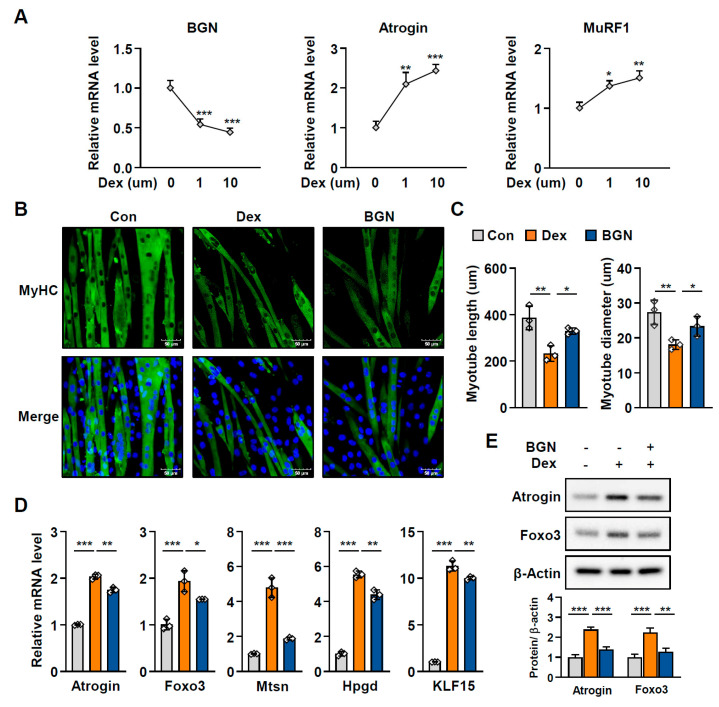
BGN alleviates Dex-induced muscle atrophy in C2C12 myotubes. (**A**) qPCR analysis of *BGN*, *Atrogin*, and *MuRF1* expression relative to that of *GAPDH* in C2C12 cells treated with 1 µM or 10 µM dexamethasone (Dex) or ethanol (as a negative control) for 24 h. (**B**) Representative immunofluorescence staining of MyHC in C2C12 myotubes treated with 10 µM Dex for 24 h or pretreated with BGN (1.0 μg/mL) before the addition of Dex. MyHC protein is shown in green, and nuclei are shown in blue (DAPI). Scale bars = 50 µm. (**C**) Quantification of myotube length and diameter. (**D**) qPCR analysis of genes related to protein degradation in myotubes (normalized to *GAPDH*). (**E**) Western blot analysis of atrogin and Foxo3 expression in myotubes and the corresponding quantification. The data are presented as the mean  ±  SD; * *p* < 0.05, ** *p* < 0.01, and *** *p* < 0.001.

**Figure 4 ijms-26-08286-f004:**
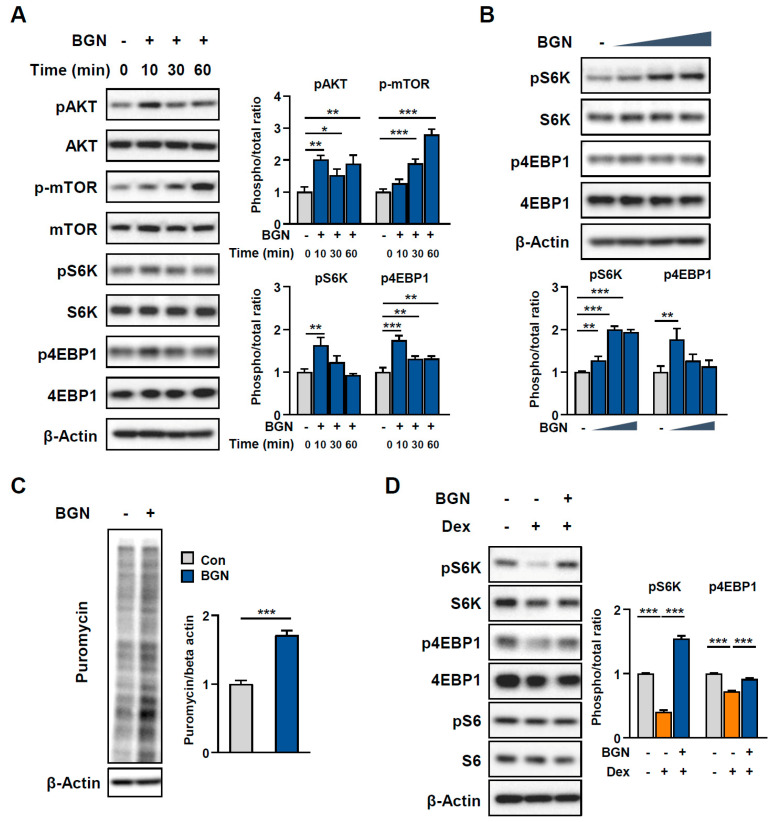
BGN improves protein synthesis by activating the AKT/mTOR pathway. (**A**) Western blot analysis and quantification of total and phosphorylated AKT, mTOR, pS6K, and p4EBP1 in C2C12 myotubes treated with 0.5 µg/mL BGN or PBS (as a negative control). (**B**) Western blot analysis and quantification of total and phosphorylated S6K and 4EBP1 in C2C12 myotubes treated with BGN (0.5, 1.0, or 1.5 μg/mL) or PBS (as a negative control) for 48 h. (**C**) Representative immunoblot of puromycin incorporation in C2C12 myotubes and the corresponding quantification. (**D**) Western blot analysis and quantification of total and phosphorylated S6K, 4EBP1, and S6 in C2C12 myotubes treated with 10 µM Dex with or without 0.5 µg/mL BGN pretreatment for 24 h. The data are presented as the mean  ±  SD; * *p* < 0.05, ** *p* < 0.01, and *** *p* < 0.001.

**Figure 5 ijms-26-08286-f005:**
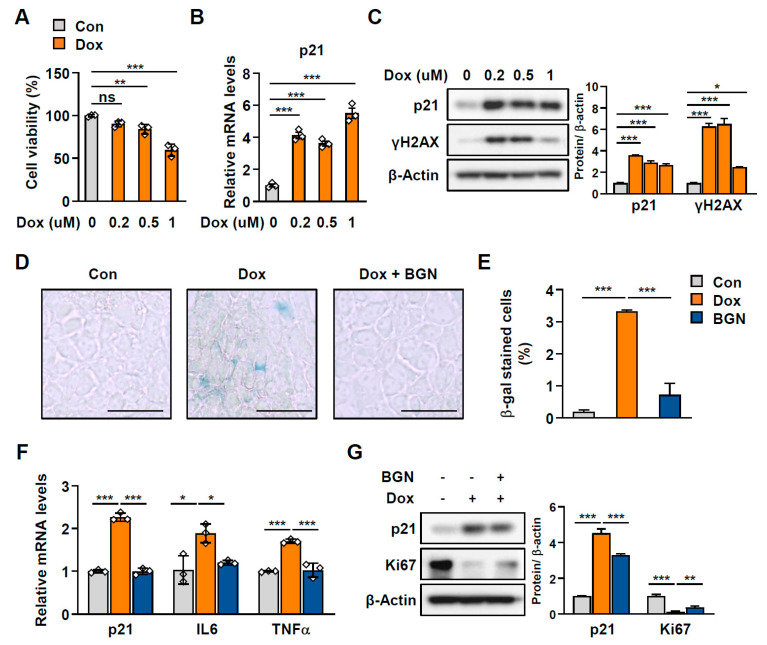
BGN mitigates cellular senescence in hepatocytes. (**A**) Viability of AML-12 hepatocytes treated with 0, 0.2, 0.5, or 1 µM Dox for 24 h. PBS was used as a negative control. (**B**) Relative mRNA expression of *p21* in hepatocytes treated with Dox (normalized to *L32* mRNA). (**C**) Western blot of p21 and γH2AX in hepatocytes treated with Dox and the corresponding quantification. (**D**) Representative images of senescence-associated β-galactosidase (SA-β-gal) staining of hepatocytes treated with 0.2 µM Dox for 24 h with or without 1 µg/mL BGN pretreatment. Scale bars = 50 µm. (**E**) Quantification of SA-β-gal-positive cells. (**F**) Relative mRNA expression of *p21*, *IL6*, and *TNFα* in hepatocytes (normalized to *L32* mRNA). (**G**) Western blot of p21 and Ki67 in hepatocytes and the corresponding quantification. The data are presented as the mean  ±  SD; * *p* < 0.05, ** *p* < 0.01, and *** *p* < 0.001.

**Figure 6 ijms-26-08286-f006:**
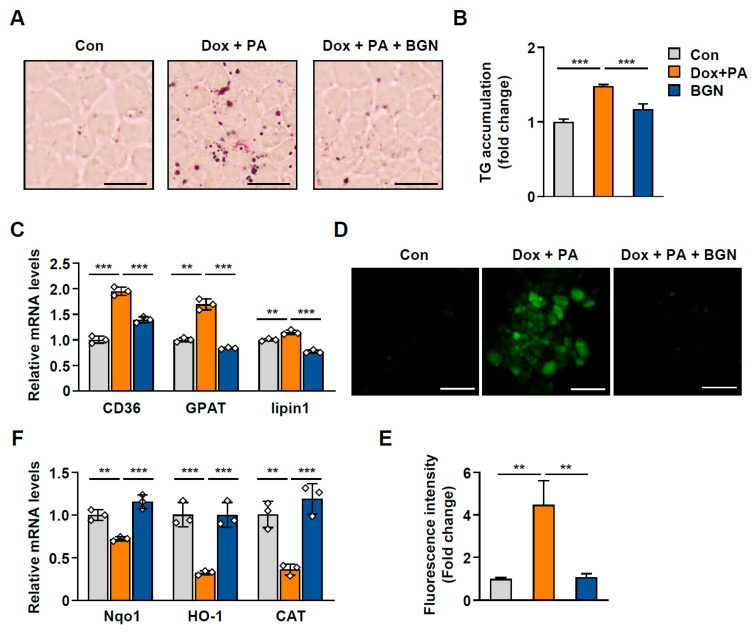
BGN attenuates senescence-driven triglyceride accumulation in hepatocytes. (**A**–**F**) AML12 hepatocytes were treated with 0.2 µM Dox and 200 µM PA (palmitic acid) for 24 h with or without 1 μg/mL BGN pretreatment. BSA (2%) was used as a negative control. (**A**) Representative Oil Red O staining. Scale bars = 25 µm. (**B**) Relative TG levels. (**C**) Relative mRNA expression of *CD36*, *GPAT*, and *lipin1* in hepatocytes (normalized to *L32* mRNA). (**D**) Representative fluorescence images of intracellular ROS. Scale bars = 100 µm. (**E**) Relative fluorescence intensity. (**F**) Relative mRNA expression of Nqo1, HO-1, and catalase-1 in hepatocytes (normalized to *L32* mRNA). The data are presented as the mean  ±  SD; ** *p* < 0.01, and *** *p* < 0.001.

**Figure 7 ijms-26-08286-f007:**
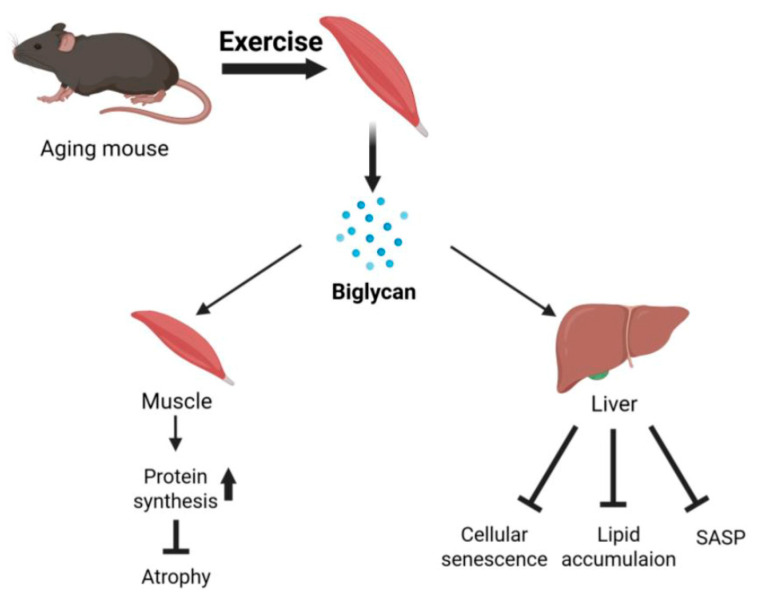
Schematic representation of our proposed mechanism. Graphical diagram (created using BioRender.com) showing the regulatory roles of BGN in aging-associated signaling pathways in skeletal muscle and liver.

## Data Availability

The datasets analyzed in the study are available from the corresponding author upon reasonable request.

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
