# Peer review of "Biglycan Alleviates Age-Related Muscle Atrophy and Hepatocellular Senescence"

_ijms, 2025, doi:10.3390/ijms26178286_

Round 1

Reviewer 1 Report

Comments and Suggestions for Authors

In this manuscript, Lee et al. provide evidence for the function of biglycan (BGN) as a novel myokine using both mouse models and cell line data. The authors demonstrate that BGN activates protein synthesis and the AKT/mTOR signaling pathway in C2C12 cells while suppressing doxorubicin-induced senescence in AML12 cells. While the study presents interesting findings, several points require clarification to strengthen the manuscript.

Major Comments

  1. Discrepancy in Myokine Classification

In the introduction, the authors claim that "Hartwig et al. identified 305 proteins classified as potential myokines from the secretome of primary human skeletal muscle cells (hSkMCs), of which BGN was included as a myokine with a signal peptide secreted by hSkMCs [9]." However, upon examination of Table 4 in Reference 9, biglycan is not listed among the 12 potential novel myokines containing a secretory signal peptide. If biglycan contains non-classical signal peptides, additional information is required, such as a supplementary figure showing the protein domain structure of BGN. Please specify the original source where BGN was identified as a myokine and clarify this discrepancy.

  1. Recombinant BGN Protein Details

The methods section lacks essential information regarding the source and preparation of recombinant BGN proteins used in the BGN treatment experiments. Please provide detailed information about the commercial source, catalog number, preparation protocol, and quality control measures for the recombinant BGN protein.

  1. Statistical Reporting

The authors state that "All the results are expressed as means ± standard errors of the means (SEMs)." However, standard deviation (SD) is generally more appropriate for describing data variability in experimental studies. Please justify the use of SEM or consider reporting SD instead.

  1. BGN Expression Inconsistency

Figure 2B shows that BGN protein levels are nearly absent at day 0 compared to day 6, suggesting minimal BGN expression in myoblasts. However, Figure 2C demonstrates relatively high BGN levels in myoblast medium (~0.4 ng/ml) compared to myotube medium (~0.6 ng/ml). What is the possible reason underlying this observation?

  1. Experimental Design for Dexamethasone Studies

In Figure 3, which shows "BGN alleviates Dex-induced muscle atrophy in C2C12 myotubes," it is unclear whether BGN was administered simultaneously with dexamethasone. If co-administered, there should be a temporal gap between treatments to avoid potential direct interactions between BGN and dexamethasone that could confound the therapeutic effects. Additionally, the qPCR results in Fig. 3D do not demonstrate a clear dose-dependent effect of BGN (except for Atrogin in the Western blot results in Fig. 3E). Please explain this discrepancy and consider revising the experimental design.

Minor Comments

  1. Statistical Analysis and Data Presentation

In the results section, the authors reference BGN levels in human plasma that gradually reduce with age based on proteomic data from Lehallier et al. (Figure S1). In Supplementary Figure S1B, the line graph presentation is inappropriate since the subjects aged 21-107 years are not the same individuals followed longitudinally. A scatter plot would be more suitable. Additionally, please specify the statistical method used (Pearson or Spearman correlation), provide the correlation coefficient (effect size), as well as p-value.

  1. Normalization Standards

The methods section states that "All the data were normalized to the expression of GAPDH or Hprt." Please specify which normalization standard (GAPDH or Hprt) was used for each figure to ensure transparency and reproducibility.

  1. Units and Quantification

In Figure 1A, the term "Relative mRNA levels" should include appropriate units such as TPM, FPKM, or normalized counts since it is not qPCR result. Furthermore, since Figure 2 presents BGN levels in absolute units (ng/ml), the plasma BGN levels in Figure 1G should also provide absolute values rather than fold changes for consistency.

  1. Discussion Clarification

In line 291 of the discussion, the authors state: "These results support the hypothesis that exercise-induced BGN may drive hepatic aging and metabolic functions through the muscle–liver axis." This statement suggests that BGN promotes hepatic aging, when the data actually suggest that BGN delays or ameliorates aging. Please revise this statement for accuracy.

  1. Rationale for Hepatocyte Studies

Please provide the rationale for testing biglycan effects specifically in hepatocytes. Were other cell types investigated?

Author Response

Q1) Discrepancy in Myokine Classification. Please specify the original source where BGN was identified as a myokine and clarify this discrepancy.
Author response: The 12 proteins secreted by human skeletal muscle cells, listed in Table 4 of Reference 9, were not previously described as myokines. Although it was not on the list of 12 proteins, BGN, as secretory proteins with signal peptide, was included in the 305 proteins released from muscle cells and listed in supplementary Table 1. Since BGN has previously been identified as a soluble secretome of human muscle cells [1], it was not included among the newly identified factors discussed in this study. However, to describe BGN identified in a comprehensive secretome profiling analysis of differentiated primary human skeletal muscle cells (hSkMC) derived from healthy adult donors, we have referred to the results of Hartwig et al. Of course, according to reviewer suggestion, we have added the protein domain structure of BGN to the supplementary Figure 1. 
[1] Le Bihan MC, Bigot A, Jensen SS, Dennis JL, Rogowska-Wrzesinska A, Lainé J, Gache V, Furling D, Jensen ON, Voit T, Mouly V, Coulton GR, Butler-Browne G. In-depth analysis of the secretome identifies three major independent secretory pathways in differentiating human myoblasts. J Proteomics. 2012 Dec 21;77:344-56. doi: 10.1016/j.jprot.2012.09.008.
Q2) Recombinant BGN Protein Details. Please provide detailed information about the commercial source, catalog number, preparation protocol, and quality control measures for the recombinant BGN protein.
Author response: Following the reviewer’s comments, we have included details of recombinant BGN protein in the cell culture and reagent section of Materials and Methods.

Q3) The authors state that "All the results are expressed as means ± standard errors of the means (SEMs)." However, standard deviation (SD) is generally more appropriate for describing data variability in experimental studies. Please justify the use of SEM or consider reporting SD instead.
Author response: Thank you for this suggestion. As suggested by the reviewer, we have calculated the standard deviation (SD) and revised the statistical analysis section in materials and methods as well as the Figures and Figure legends.

Q4) BGN Expression Inconsistency. Figure 2B shows that BGN protein levels are nearly absent at day 0 compared to day 6, suggesting minimal BGN expression in myoblasts. However, Figure 2C demonstrates relatively high BGN levels in myoblast medium (~0.4 ng/ml) compared to myotube medium (~0.6 ng/ml). What is the possible reason underlying this observation?
Author response: We agree that this is an important consideration. First, we reviewed experimental procedures to measure proteins secreted from cells. In our experiment, the media was replaced with serum-containing media before and after C2C12 cell differentiation. After 48 hours of incubation, the medium was collected and analyzed by ELISA. However, after an extensive literature review, we found that the highest protein secretion occurs approximately 1 hour after fresh media replacement [1]. Thus, we modified our protocol to minimize nonspecific protein release, enabling more precise measurement of myokines secreted from cells. Based on this insight, the cell culture medium was changed with fresh DMEM (serum-free) both before (myoblasts) and after (myotube) differentiation, and collected the medium 1 hour post-replacement for BGN measurement. Using this optimized method, we observed that BGN secretion increased by more than fourfold following myotube compared to myoblasts levels. Consequently, this data is now presented in Figure 2C and the measurement protocol was revised accordingly in the Methods section. In addition, we measured BGN secretion following forskolin treatment using the same protocol and accordingly revised Figure 2E.
[1] Furuichi Y, Manabe Y, Kakagi M, Aoki M, Fujii NL. Evidence for acute contraction-induced myokine secretion by C2C12 myotubes. PloS one. 2018 Oct 24;13(10): e0206146. doi:10.1371/journal.pone.0206146

Q5) Experimental Design for Dexamethasone Studies. In Figure 3, which shows "BGN alleviates Dex-induced muscle atrophy in C2C12 myotubes," it is unclear whether BGN was administered simultaneously with dexamethasone. If co-administered, there should be a temporal gap between treatments to avoid potential direct interactions between BGN and dexamethasone that could confound the therapeutic effects. Additionally, the qPCR results in Fig. 3D do not demonstrate a clear dose-dependent effect of BGN (except for Atrogin in the Western blot results in Fig. 3E). Please explain this discrepancy and consider revising the experimental design.
Author response: In experiments, C2C12 myotubes were pretreated with recombinant BGN for 1-2h and then incubated with dexamethasone or vehicle (ethanol) for 24h. We have added this procedure in Figure legends.
We agree with reviewer’s comment. Accordingly, we have revised the Fig. 3B-3E based on the experimental results for myotube length following pretreatment with 1ug/ml of BGN

Minor Comments
Q1) In the results section, the authors reference BGN levels in human plasma that gradually reduce with age based on proteomic data from Lehallier et al. (Figure S1). In Supplementary Figure S1B, the line graph presentation is inappropriate since the subjects aged 21-107 years are not the same individuals followed longitudinally. A scatter plot would be more suitable. Additionally, please specify the statistical method used (Pearson or Spearman correlation), provide the correlation coefficient (effect size), as well as p-value.
Author response: Thank you for pointing this out. As suggested by the reviewer, we have revised it.

Q2) The methods section states that "All the data were normalized to the expression of GAPDH or Hprt." Please specify which normalization standard (GAPDH or Hprt) was used for each figure to ensure transparency and reproducibility.
Author response: Thank you for this suggestion. We have presented standard reference genes used in the RT-qPCR data to each Figure legend.

Q3) In Figure 1A, the term "Relative mRNA levels" should include appropriate units such as TPM, FPKM, or normalized counts since it is not qPCR result. Furthermore, since Figure 2 presents BGN levels in absolute units (ng/ml), the plasma BGN levels in Figure 1G should also provide absolute values rather than fold changes for consistency.
Author response: Thank you for pointing this out. As suggested by the reviewer, we have revised Figure 1A and Figure 1G.

Q4) In line 291 of the discussion, the authors state: "These results support the hypothesis that exercise-induced BGN may drive hepatic aging and metabolic functions through the muscle–liver axis." This statement suggests that BGN promotes hepatic aging, when the data actually suggest that BGN delays or ameliorates aging. Please revise this statement for accuracy.
Author response: We thank the reviewer for the constructive suggestion. As suggested by the reviewer, we have revised the sentence in the discussion section as follow:
These results support the hypothesis that exercised-induced BGN exert effects that counteract oxidative stress and fat accumulation-induced hepatocyte senescence via myokine-mediated skeletal muscle-liver crosstalk.

Q5) Please provide the rationale for testing biglycan effects specifically in hepatocytes. Were other cell types investigated?
Author response: The prevalence of non-alcoholic fatty liver disease (NAFLD) is increasing with the rising elderly population and has emerged as a growing global public health issue due to increased life expectancy. Notably, NAFLD is known to elevate the risk of both hepatic and cardiovascular-related mortality. In our previous study, we demonstrated that vitamin D attenuates age-related muscle atrophy, mimicking the beneficial effects of exercise. Furthermore, we found that vitamin D supplementation prevented aging-associated NAFLD development in aged mice. Importantly, we observed that vitamin D enhances the secretion of myokines, among which BGN was notably upregulated, similar to the effects of exercise, as shown in our preliminary data. Based on these findings, we aimed to investigate whether BGN secreted from skeletal muscle could alleviate hepatocyte senescence and lipid accumulation. In addition, given the metabolic importance and heightened susceptibility to aging of white adipose tissue (WAT), we plan to explore the effects of BGN on WAT in future studies.

Reviewer 2 Report

Comments and Suggestions for Authors

Dear Authors,

I have carefully reviewed your manuscript and recommend that you address an important topic related to age-related muscle atrophy and hepatocellular senescence through the investigation of biglycan (BGN) as a potential modulator.

Your findings are promising and may contribute significantly to the field. However, I have identified several areas that require revision or clarification before the manuscript can be considered for publication:

  1. References – Formatting Inconsistencies
  • Please ensure all references follow a uniform format. Currently, there is an inconsistency in:
    • Journal titles: Some are abbreviated (e.g., Faseb j, Mol Metab) while others are in full (e.g., Nature Communications).
    • Use of capital letters in article titles: Some titles capitalize only the first word, while others capitalize each word (e.g., Ref 10).
    • Page ranges: Certain references lack page numbers (e.g., Refs 10 and 23), which should be included if available.

  1. Language – Minor English Issues

The manuscript is generally clear, but a few minor grammatical issues should be addressed through careful proofreading. For example:

  • In the Abstract, the phrase "suggest beneficial roles of BGN" should be corrected to

"suggest beneficial roles for BGN"

I recommend a thorough language revision to enhance clarity throughout the manuscript.

  1. Materials and Methods – Clarifications Needed

The methodology section would benefit from additional details and justifications to improve reproducibility and scientific rigor:

  • Doxorubicin-induced aging model: Please provide a reference or explanation supporting the validity of this model as a proxy for age-related changes. What evidence supports its use in this context?
  • Exercise intervention: Clarify the intensity, duration, and treadmill settings used (e.g., speed, incline), as these parameters are critical for interpreting the physiological impact of the intervention.
  • Experimental controls: Were negative or positive controls included for Dox and/or BGN treatments in AML12 or C2C12 cells? Including this information would help assess the specificity and robustness of the observed effects.
  • Sample size (n): The number of animals per group and the number of biological or technical replicates for each experiment are not reported. This information is essential for the evaluation of statistical validity and reproducibility.

I believe these revisions will significantly improve the manuscript and its clarity. I look forward to reviewing a revised version that addresses these points.

Author Response

Q1) References – Formatting Inconsistencies. Please ensure all references follow a uniform format.
Author response: We apologize for the confusion. We have revised the references a uniform format.

Q2) The manuscript is generally clear, but a few minor grammatical issues should be addressed through careful proofreading.
Author response: We had the manuscript proofread in English language editing service by AJE (American Journal Experts) prior to submission. However, in accordance with the reviewer suggestion, we have it proofread again and thoroughly reviewed it before resubmitting.

Q3) Materials and Methods – Clarifications Needed
(Q3-a) Doxorubicin-induced aging model: Please provide a reference or explanation supporting the validity of this model as a proxy for age-related changes. What evidence supports its use in this context?
Author response: Pundlik et al. [1] reported that Doxorubicin (Dox) treatment promoted senescence in mouse skeletal muscles along with elevated expression of senescence markers (p21, p16). Based on this study, we divided the mice into two groups to test the aging effects of Dox (Group 1: 5 mg/kg every 3 days for 2 weeks, Group 2: 2 mg/kg every 3 days for 4 weeks). The results showed a significant exhibition of the aging phenotype in the 5 mg/kg / 2 weeks group, and we obtained the same results in repeated experiments. Consequently, we generated Dox-induced aging mouse model and confirmed the expression of senescence markers including SASP-related factors and protein degradation markers as shown in Fig. 1E. We have added the reference to the animal experiments section of methods following the reviewer’s comment.  
[1] Pundlik SS, Barik A, Venkateshvaran A, Sahoo SS, Jaysingh MA, Math RGH, Lal H, Hashmi MA, Ramanathan A. Senescent cells inhibit mouse myoblast differentiation via the SASP-lipid 15d-PGJ2 mediated modification and control of HRas. Elife. 2024 Aug 28;13:RP95229. doi: 10.7554/eLife.95229

(Q3-b) Exercise intervention: Clarify the intensity, duration, and treadmill settings used (e.g., speed, incline), as these parameters are critical for interpreting the physiological impact of the intervention.
Author response: We have added a method of exercise intervention.

(Q3-c) Experimental controls: Were negative or positive controls included for Dox and/or BGN treatments in AML12 or C2C12 cells? Including this information would help assess the specificity and robustness of the observed effects.
Author response: We have added negative controls for all treatment experiments to the Figure legends and Materials and Methods.

(Q3-d) Sample size (n): The number of animals per group and the number of biological or technical replicates for each experiment are not reported. This information is essential for the evaluation of statistical validity and reproducibility.
Author response: We have added the number of animals and revised statistical analysis section in Materials and Methods.

Round 2

Reviewer 1 Report

Comments and Suggestions for Authors

The authors have adequately addressed the majority of my previous concerns. However, minor revisions are still required to strengthen the manuscript:
1. Clarification of Therapeutic Claims
While the authors state in the abstract (line 25) that "Our results suggest that BGN has beneficial effects against muscle atrophy and hepatocellular senescence, indicating its potential as a therapeutic target in the context of age-related diseases," the experimental design consistently employs BGN pretreatment throughout all studies presented. I would like to clarify whether the authors have conducted experiments involving BGN administration following Doxorubicin or dexamethasone treatment. If such post-treatment data are not available, the characterization of BGN as a "therapeutic target" should be qualified in the discussion section, as the current evidence demonstrates protective rather than therapeutic effects.
2. Figure 1A Y-axis Labeling
In the revised Figure 1A, the y-axis is labeled "Relative expression (FPKM)." Given that FPKM represents an already normalized value, please clarify whether this data has been further normalized (e.g., relative to a housekeeping gene). I recommend either specifying the meaning of "relative" in this context or simplifying the label to "FPKM" if no additional normalization was performed.
3. Gene Nomenclature and Additional Information
Two minor corrections are needed:
a) The gene symbol for Myostatin is incorrectly labeled as "Mtsn" (line 142, Figure 3D) and should be corrected to "MSTN."
b) The authors have included new data regarding 15-hydroxyprostaglandin dehydrogenase (Hpgd/15-PGDH). Please provide a brief description of how this protein relates to protein degradation pathways to enhance reader comprehension.

Author Response

Comments 1) While the authors state in the abstract (line 25) that "Our results suggest that BGN has beneficial effects against muscle atrophy and hepatocellular senescence, indicating its potential as a therapeutic target in the context of age-related diseases," the experimental design consistently employs BGN pretreatment throughout all studies presented. I would like to clarify whether the authors have conducted experiments involving BGN administration following Doxorubicin or dexamethasone treatment. If such post-treatment data are not available, the characterization of BGN as a "therapeutic target" should be qualified in the discussion section, as the current evidence demonstrates protective rather than therapeutic effects.
Response 1) We very much appreciate this helpful comment. To investigate the effect of BGN on muscle atrophy, we first conducted an experiment in which BGN was treated 2 hours before or 6 hours after dexamethasone treatment. As a result, we found that pre-treatment with BGN was more effective in attenuating Dex-induced muscle atrophy than post-treatment. Given the absence of such post-treatment data in this study, we fully agree with the reviewer’s comment that a “protective effect” is more appropriate than a "therapeutic target". Thus, we have revised the abstract (line 25) and discussion (line 318) to the following sentence. 

BGN has beneficial effects against muscle atrophy and hepatocellular senescence, indicating its potential as a protective factor for age-related diseases.

Comments 2) In the revised Figure 1A, the y-axis is labeled "Relative expression (FPKM)." Given that FPKM represents an already normalized value, please clarify whether this data has been further normalized (e.g., relative to a housekeeping gene). I recommend either specifying the meaning of "relative" in this context or simplifying the label to "FPKM" if no additional normalization was performed.
Response 2) We very much appreciate this helpful comment. Following your suggestion, we have corrected it to FPKM.

Comments 3-a) The gene symbol for Myostatin is incorrectly labeled as "Mtsn" (line 142, Figure 3D) and should be corrected to "MSTN."
Response 3-a) We apologize for the confusion. We have corrected the mislabeled parts (line 141, Figure 3D).

Comments 3-b) The authors have included new data regarding 15-hydroxyprostaglandin dehydrogenase (Hpgd/15-PGDH). Please provide a brief description of how this protein relates to protein degradation pathways to enhance reader comprehension.
Response 3-b) We appreciate the reviewer’s insightful comments. Palla et al. [1] identified increased expression of 15-hydroxyprostaglandin dehydrogenase (Hpgd/15-PGDH), the enzyme responsible for degrading prostaglandin-E2 (PGE2), as a hallmark of aged muscles, both in mice and humans. In line with this result, we have previously shown that the levels of Hpgd were significantly increased in the aged mice but were restored by exercise [2]. Elevated Hpgd expression in aged mice decreased PGE2 levels, and the consequent reduction in PGE2-AKT signaling led to Foxo activation and subsequent induction of E3 ubiquitin ligases (Atrogin and MuRF1), thereby enhancing protein degradation. Additionally, overexpression of Hpgd in young muscles induced atrophy. In aged mice, inhibition of 15-PGDH, both AAV-mediated Hpgd knockdown and pharmacological inhibition using a small molecule, counters muscle atrophy and markedly increased muscle mass, strength, and endurance. Furthermore, RNA-seq analysis of aged mice treated with a Hpgd inhibitor revealed downregulation of the ubiquitin-proteasome pathway in muscle. Namely, 15-PGDH inhibition reveals reduced expression of atrogenes (Atrogin, MuRF1, Traf6 as well as Mstn). 
[1] Palla AR, Ravichandran M, Wang YX, Alexandrova L, Yang AV, Kraft P, Holbrook CA, Schürch CM, Ho ATV, Blau HM. Inhibition of prostaglandin-degrading enzyme 15-PGDH rejuvenates aged muscle mass and strength. Science. 2021 Jan 29;371(6528):eabc8059. doi: 10.1126/science.abc8059.
[2] Lee, Y.J.; Kim, G.H.; Lee, D.S.; Jeong, H.J.; Lim, J.H. Activation of the Apelin/APJ System by Vitamin D Attenuates Age-Related Muscle Atrophy. Life Sci 2024, 359, 123205, doi:10.1016/j.lfs.2024.123205.

Reviewer 2 Report

Comments and Suggestions for Authors

I have noticed an inconsistency in the description of the number of mice used. In one section of the manuscript, it is stated that 10–12 mice were used per group (control group and exercise group), but later it is mentioned that 5 mice were used as a negative control group. It is necessary to clarify whether this group represents a third independent experimental group. In addition, the authors should specify whether these mice were subjected to the same handling as the others, as the handling effect can significantly influence the outcomes in this type of experiment.

Author Response

Comments 1) I have noticed an inconsistency in the description of the number of mice used. In one section of the manuscript, it is stated that 10–12 mice were used per group (control group and exercise group), but later it is mentioned that 5 mice were used as a negative control group. It is necessary to clarify whether this group represents a third independent experimental group. In addition, the authors should specify whether these mice were subjected to the same handling as the others, as the handling effect can significantly influence the outcomes in this type of experiment.
Response 1) We apologize for the confusion. In this study, we conducted two types of animal experiments. First, we utilized an aging mouse model (18-month-old) established in our previous study [1], as described in line 326-331 of Method section. These mice were divided into a control group (without any exercise intervention) and an exercise group (n=10-12 per group), which underwent an exercise intervention for four months. We then examined the expression of BGN in the skeletal muscles of these aging mice. Second, to validate the reduced expression of BGN in aged muscle, we performed an additional experiment using a Dox-induced aging model, as described in line 331-338. In this experiment, the negative control group was administered PBS, while the experimental group was treated with Dox. Each group consisted of 5 mice. The statement in the Methods section indicating that "5 mice were used as a negative control group" refers specifically to the negative control group in this second animal experiment. The number of animals used in each experiment was indicated in Figure legends.

[1] Lee, Y.J.; Kim, G.H.; Lee, D.S.; Jeong, H.J.; Lim, J.H. Activation of the Apelin/APJ System by Vitamin D Attenuates Age-Related Muscle Atrophy. Life Sci 2024, 359, 123205, doi:10.1016/j.lfs.2024.123205.